# A Viral Long Non-Coding RNA Protects against Cell Death during Human Cytomegalovirus Infection of CD14+ Monocytes

**DOI:** 10.3390/v14020246

**Published:** 2022-01-26

**Authors:** Marianne R. Perera, Kathryn L. Roche, Eain A. Murphy, John H. Sinclair

**Affiliations:** 1Cambridge Institute of Therapeutic Immunology and Infectious Disease, Department of Medicine, University of Cambridge, Addenbrooke’s Hospital, Hills Road, Cambridge CB2 0QQ, UK; mp704@cam.ac.uk; 2Evrys Bio, Pennsylvania Biotechnology Center, Doylestown, PA 18902, USA; kate@evrysbio.com; 3Microbiology and Immunology Department, SUNY Upstate Medical University, Syracuse, NY 13210, USA; murphye1@upstate.edu

**Keywords:** human cytomegalovirus, latency, long non-coding RNA, reactive oxygen species, apoptosis, oxidative stress

## Abstract

Long non-coding RNA β2.7 is the most highly transcribed viral gene during latent human cytomegalovirus (HCMV) infection. However, as yet, no function has ever been ascribed to β2.7 during HCMV latency. Here we show that β2.7 protects against apoptosis induced by high levels of reactive oxygen species (ROS) in infected monocytes, which routinely support latent HCMV infection. Monocytes infected with a wild-type (WT) virus, but not virus deleted for the β2.7 gene (Δβ2.7), are protected against mitochondrial stress and subsequent apoptosis. Protected monocytes display lower levels of ROS and additionally, stress-induced death in the absence of β2.7 can be reversed by an antioxidant which reduces ROS levels. Furthermore, we show that infection with WT but not Δβ2.7 virus results in strong upregulation of a cellular antioxidant enzyme, superoxide dismutase 2 (SOD2) in CD14+ monocytes. These observations identify a role for the β2.7 viral transcript, the most abundantly expressed viral RNA during latency but for which no latency-associated function has ever been ascribed, and demonstrate a novel way in which HCMV protects infected monocytes from pro-death signals to optimise latent carriage.

## 1. Introduction

Human cytomegalovirus (HCMV) is a widespread beta-herpesvirus that infects 40–100% of populations worldwide [1]. After primary infection, the virus can establish a latent lifecycle in cells of the early myeloid lineage, where it persists for the lifetime of the host. A hallmark of latent infection is the carriage of viral genomes in the absence of the production of infectious virions in a form that is able to reactivate and resume a full lytic infection. One important site of HCMV latency is in undifferentiated myeloid cells, such as CD34+ progenitor cells and their derivative CD14+ monocytes [2,3,4,5,6]. During latent carriage, latency-associated gene expression is known to have profound effects on the latent cell and its environment to ensure virus survival and dissemination (reviewed in [7]).

It is becoming increasingly clear that latency-associated gene expression is more complex than first thought [8,9,10]. Transcriptome analyses of monocytes and CD34+ve progenitor cells have demonstrated that the most highly transcribed viral gene during HCMV latency is the β2.7 long non-coding RNA (also known as RNA2.7) [10,11]. Long non-coding RNAs (lncRNAs) are defined as RNAs over 200 nucleotides in length with little or no protein-coding potential, but with diverse roles, which can range from acting as scaffolds for the assembly of protein complexes, to functional ‘microRNA sponges’ where they bind and sequester miRNAs away from their targets (reviewed in [12]). However, the function/s of β2.7 during latency remains unclear.

During a lytic infection, where β2.7 is also the most highly transcribed viral gene, accounting for approximately 46.8% of polyadenylated viral RNA at 72 h post infection (h.p.i) [13], it is known to protect lytically infected cells from apoptosis induced by rotenone by its interaction with mitochondrial Complex I, but the mechanism of how this protection occurs has not yet been identified [14]. 

Consequently, we have analysed the role of β2.7 expression in monocytes, in which HCMV is known to undergo a latent infection. We find that β2.7 does indeed protect infected monocytes from apoptosis and that this is mediated by preventing the accumulation of high levels of ROS, likely at least in part via upregulation of superoxide dismutase 2 (SOD2). 

## 2. Materials and Methods

### 2.1. Cells

Primary CD14+ monocytes were isolated from the peripheral blood of healthy donors or from apheresis cones (NHS Blood & Transport Service, Cambridge, UK) as described in Poole et al. [15]. In brief, peripheral blood mononuclear cells (PBMC) were separated from whole blood by density-gradient centrifugation using Lymphoprep™ (Stem Cell Technologies, Vancouver, Canada), and monocytes were isolated from PBMC by magnetic-activated cell sorting (MACS) with CD14+ microbeads (Miltenyi Biotech, Bergisch Gladbach, Germany). Monocytes were cultured in X-Vivo15 media (Lonza, Basel, Switzerland) at 37 °C in a 5% CO_2_ atmosphere. Where stated, monocytes were treated with 50 μM cadmium chloride (Sigma, St Louis, MO, USA), N-acetyl cysteine (A9165, Sigma) or MitoTEMPO (Item no. 16,621, Cayman Chemical, Ann Arbor, MI, USA). 

### 2.2. Human Cytomegaloviruses

WT Toledo and Δβ2.7 Toledo viruses were kind gifts from Gavin Wilkinson and are described in McSharry et al. [16]. WT TB40/E-SV40-GFP, also known as TB40/E-eGFP, is described in [17], and was used to generate Δβ2.7 TB40/E by standard BAC recombineering protocols detailed in [18,19,20], using the primers listed in Table 1. 

In brief, the sequence from nt 35,220 up to the TATA box at nt 37,726 within the TB40/E BAC4 sequence listed in EF999921 was deleted by homologous recombination with the galK gene. The galK gene was subsequently removed from the BAC, again by homologous recombination, using the reversion primers listed in Table 1. All recombinant viruses were validated for deletion of β2.7 by genome PCR analysis and sequencing across the deletion site.

Viruses were propagated by inoculating HFFs at an MOI of 0.1. At 90% infection, the supernatant was harvested every 2–3 days and centrifuged for 10 min at 1500× *g*, before being frozen at −80 °C. To concentrate virus, supernatants were defrosted and spun for 2 h at 18 °C at 14,500× *g* in an Avanti-J25 centrifuge with a JLA-16.250 rotor (Beckman Coulter). The pellet was resuspended in X-Vivo 15 media, aliquoted, and then stored at −80 °C. Prior to use, concentrated virus was defrosted and spun for 2 min at 800× *g* to further remove any cellular debris.

Monocytes were inoculated with virus at a multiplicity of infection (MOI) of 3 as based on titration on RPE-1 cells, and incubated for 3 h, before being washed twice with PBS and given fresh X-Vivo15 media. 

Ultra-violet light (UV) inactivation of virus was conducted by placing samples in a tissue culture plate within 10 cm of a UV germicidal (254 nm) lamp for 30 mins as described in [21]. 

### 2.3. Fluorescence Microscopy

Mitochondrial membrane potential was assayed using tetramethylrhodamine, ethyl ester (TMRE, Abcam). Cells were stained with 50 nM TMRE (Abcam, ab113852) and Hoechst 33,342 stain (Sigma, B2261) for 30 min at 37 °C in the dark, prior to being washed once with PBS and imaged on a widefield Nikon TE200 microscope. Percentage of TMRE-stained cells was determined from 3 fields of view per replicate over at least 3 independent biological repeats. 

Superoxide was stained with MitoSOX™ Red (ThermoFisher, Waltham, MA, USA). MitoSOX reagent was added to cells to a final concentration of 1 μM for 10 min at 37 °C in the dark. Cells were then washed three times with warm Hank’s buffered saline solution (HBSS, Sigma), Hoeschst stained and imaged on a widefield Nikon TE200 microscope. Quantification of MitoSOX fluorescence was performed using ImageJ software, analysing a minimum of 400 cells per replicate. 

### 2.4. Sodium Dodecyl Sulphate-Polyacrylamide Gel Electrophoresis (SDS-PAGE) and Western Blotting

Samples were directly lysed in Laemmli Buffer and separated by SDS-PAGE on a 12% polyacrylamide gel. Separated proteins were transferred to Hybond nitrocellulose membranes (Amersham Biosciences, Amersham, UK) and blocked in 5% milk in Tris-buffered saline (milk-TBS). Primary antibodies were diluted in milk-TBS as follows: cleaved caspase-3 (CST, #9664, 1/500), PARP (CST, #9542, 1/1000), β-actin (Abcam, ab8227, 1/2000), SOD2 (Abcam, ab13533, 1/1000), and GPX-1 (Abcam, ab22604, 1/1000), and incubated with membranes overnight at 4 °C. Secondary antibody (chicken anti-rabbit-HRP, Abcam, ab6829, 1/2000) was incubated with membrane for 1 hr at room temperature. Blots were incubated with Amersham ECL Western Blotting detection reagent (GE Healthcare Life Sciences) for 5 min and then exposed to autoradiography film (Fujifilm, Tokyo, Japan). All densitometry analysis was performed using ImageJ software. 

## 3. Results

### 3.1. β2.7 Protects against Mitochondrial Stress in Infected Monocytes

To test if β2.7 has a protective role during latent infection, we stressed infected monocytes with cadmium chloride (CdCl_2_), which is an established way of causing apoptosis in primary lymphocytes and monocytes by inducing loss of mitochondrial membrane potential (MMP) [22,23]; similar effects have also been demonstrated in monocytic U937 and THP-1 cell lines [24,25]. The exact mechanism by which cadmium causes loss of MMP and cytotoxicity in monocytic cells is unclear: some reports suggest cadmium activates caspase-8 [24] while others suggest its similarity in diameter and charge to calcium allows it to competitively inhibit calcium influx [24]. However, multiple studies have indicated that cadmium increases ROS levels [22,25,26]. Consequently, we tested the effect of cadmium ions in infected monocytes in the presence or absence of β2.7 gene expression.

We started by analysing mitochondrial function in infected monocytes using the fluorescent dye, TMRE, a positively charged, cell permeant molecule which accumulates in the mitochondrial matrix of actively respiring mitochondria. It therefore serves as a measure of MMP, such that mitochondrial oxidative phosphorylation uncouplers such as FCCP, which disrupts ATP synthesis by depolarising mitochondrial membrane potential, cause dye dispersal and loss of TMRE fluorescence (Figure 1A–C, and reviewed in [27]). 

To investigate the protective function of β2.7 during latent infection, we mock infected, or infected CD14+ monocytes with wild type (WT) virus or virus with a β2.7 gene deletion (Δβ2.7) in the Toledo strain of HCMV. Monocytes infected with HCMV are known to undergo latent infection which can be maintained in long-term culture [5,11,28,29,30,31,32] and the Toledo strain of HCMV has been shown to undergo latent infection in myeloid progenitor cells [33,34,35,36,37]. Fourteen days post infection (d.p.i), cells were either treated or mock-treated with cadmium chloride for 24 h, and then TMRE and Hoechst stained. As expected, the addition of cadmium chloride to mock infected cells caused a severe drop in the number of cells with visible TMRE staining, indicating mitochondrial depolarization (Figure 2A,B). By contrast, monocytes infected with WT virus were partially protected from this cadmium-induced loss of TMRE staining, but this protection was lost in the absence of β2.7 (Figure 2A,B). This strongly suggests that β2.7 has a protective role against mitochondrial stress in latent infection. 

To ensure this effect was not virus isolate-specific and to specifically analyse infected cells in the cell population (which could not be tracked in untagged WT Toledo virus infections), we repeated this analysis at earlier timepoints (3 d.p.i and 5 d.p.i) with GFP-tagged WT or GFP-tagged Δβ2.7 TB40/E viruses (Figure 2C,D). WT infected monocytes consistently retained their TMRE staining when treated with cadmium chloride, but this protection was lost when cells were infected with Δβ2.7 virus, again indicating that β2.7 protects cells from mitochondrial stress during latency. 

### 3.2. β2.7 Protects against Apoptosis in HCMV Infected Monocytes

A loss in MMP is known to either stimulate apoptosis or be a consequence of it [38]. Therefore, we next tested whether the cadmium-induced loss of MMP in monocytes was accompanied by apoptosis and whether this was impacted upon by latent infection. Primary CD14+ monocytes were isolated and either mock infected or infected with WT or Δβ2.7 Toledo virus. At 6 d.p.i, cells were treated or mock treated with CdCl_2_ and 24 h post treatment, protein was harvested and assayed for levels of cleaved caspase-3, a hallmark of apoptosis (Figure 3). Addition of CdCl_2_ to mock infected cells resulted in detectable cleaved caspase-3 levels which was not observed in WT virus-infected cells treated with cadmium (Figure 3). By contrast, cadmium-treated Δβ2.7-infected cells showed a dramatic increase in cleaved caspase-3. Taken together, these results suggest that cadmium treatment of monocytes results in low, but discernible induction of apoptotic markers, which is prevented by infection with WT virus. In contrast, this protection is lost in the absence of β2.7 and, indeed, infection with Δβ2.7 virus results in a massively increased level of cleaved caspase-3, suggesting that infection itself substantially stresses the cells but the presence of β2.7 reverses this phenotype. Therefore, viral β2.7 appears to be important in protecting infected monocytes from stress-induced apoptosis. 

### 3.3. β2.7 Lowers ROS Levels in Infected Monocytes

HCMV infection protects monocytes from cadmium-induced cell death (Figure 2 and Figure 3). As cadmium is an inducer of ROS in monocytic cells [22,25], and the β2.7 gene in the absence of other HCMV factors, lowers ROS levels when over-expressed in rat aortic endothelial cells [39], we reasoned that β2.7 might lower ROS levels in the context of a latent infection. ROS are formed when oxygen accepts an extra electron to produce a highly reactive molecule. This includes free radicals such as superoxide (O_2_*-) and hydroxyl radicals (HO*-), as well as their derivatives, e.g., hydrogen peroxide (H_2_O_2_), peroxynitrite (ONOO-), and hypochlorous acid (HOCl) [40]. At high levels (known as oxidative stress), they react with and damage, proteins, DNA, and lipids, ultimately leading to apoptosis and/or necrosis [41,42]. The majority of ROS production in the cell can be traced to complex I and complex III of the electron transport chain in the mitochondria; though in phagocytic cells, they are also generated in the extracellular fluid or interior of phagosomes by NADPH oxidase (NOX) [43].

To test whether β2.7 modulates ROS levels during a latent infection, we mock infected or infected CD14+ monocytes with WT or Δβ2.7 Toledo virus for 6 days and then treated them with cadmium chloride. Twenty-four hours post treatment, cells were stained with Hoechst and MitoSOX, a mitochondrial superoxide dye (Figure 4). 

Consistent with β2.7 protecting cells from superoxide induction during infection, we observed increased MitoSOX staining in Δβ2.7 infected cells even in the absence of exogenous ROS induction, indicating that infection in the absence of β2.7 increases ROS levels (Figure 4A). This difference became substantially more pronounced when cells were also stressed with cadmium chloride: exogenously stressed cells infected with Δβ2.7 Toledo virus displayed much higher levels of superoxide than WT infected cells (Figure 4B,C). Interestingly, protection from high ROS levels seemed to occur in all cells observed rather than a subset of infected cells. Protection of bystander cells could be mediated by a secreted factor, which we further address in Section 3.5. 

### 3.4. Antioxidants or β2.7 Reduce ROS Levels Induced by Cadmium to Prevent Apoptosis

Although our observations clearly showed that β2.7 expression results in lower ROS levels, as well as protection from apoptosis in exogenously stressed cells, it was important to determine whether the high ROS levels we observed in the absence of β2.7 were driving cell death rather than simply being a side effect of it. To address this, we first tested whether ROS were causing mitochondrial stress by adding cadmium to CD14+ monocytes following treatment with increasing concentrations of antioxidants to lower ROS. We then measured MMP to test whether lowering ROS levels with an antioxidant would rescue mitochondrial activity. As can be seen in Figure 5A,B, the use of two different antioxidants, either N-acetyl cysteine (NAC) or MitoTEMPO, prevented the loss of MMP induced by cadmium ions in CD14+ monocytes. This suggests that high ROS levels induced by cadmium are indeed responsible for the loss in MMP and are not simply a by-product of cell stress. 

We then wished to investigate whether the antioxidant, NAC, would also rescue Δβ2.7 infected cells from cadmium-induced apoptosis. We treated mock, WT or Δβ2.7 virus-infected monocytes with cadmium, with or without NAC, and immunoblotted protein lysates from these cells for cleaved caspase-3 and cleaved PARP (both indicative of apoptosis). As expected, treatment of monocytes with cadmium induced detectable levels of cleaved caspase-3 and cleaved PARP which was substantially increased in Δβ2.7 infected cells (Figure 5C). However, the addition of NAC to Δβ2.7-infected monocytes prevented this increase in cleaved caspase-3 or cleaved PARP, consistent with high ROS levels being a major factor in inducing apoptosis in Δβ2.7 infected cells. 

### 3.5. WT, but Not Δβ2.7, Virus Infection of Monocytes Upregulates the Antioxidant Enzyme, SOD2

Our data, so far, indicated that β2.7 protects infected cells from the effects of high ROS levels; however, the mechanism for this was unclear. One important regulator of ROS is the mitochondrial enzyme superoxide dismutase 2 (SOD2) [44]. Electrons that leak from complex I and III of the electron transport chain due to e.g., metabolic stress, can reduce oxygen molecules to the highly reactive and damaging superoxide. SOD2 limits the accumulation of ROS by catalysing the conversion of superoxide to hydrogen peroxide, which is then further oxidized to water and oxygen by GPX-1, catalase or PDRX1 (see Figure 6A) [45,46]. Interestingly, SOD2 mRNA was found to be upregulated in a previous unbiased screen for altered transcripts after HCMV infection of monocytes [47]. 

We reasoned, therefore, that a possible mechanism of action of β2.7 could be through modulation of SOD2 protein levels. Consequently, we analysed cellular SOD2 levels during infection in the presence or absence of β2.7. Figure 6B shows that SOD2 is significantly upregulated in monocytes infected with WT Toledo virus but not after infection with Δβ2.7 virus. Interestingly, a time course analysis showed that this virus-induced increase in SOD2 occurred as early as 1 d.p.i and it steadily increased over time (Figure 6C), again dependent on the presence of β2.7. 

As the cellular enzyme, GPX-1, also functions in the detoxification of ROS (see Figure 6A), we also examined levels of GPX-1 protein during infection. As shown in Figure 6D, the levels of GPX-1 are also upregulated in WT but not Δβ2.7 infected monocytes, albeit not as strongly as for SOD2. Therefore, WT infection of monocytes seems to also increase levels of another antioxidant enzyme, presumably to aid in further detoxification of ROS. 

Finally, we were surprised that despite only ~10% of monocytic cells establishing a latent infection [48,49], we observed drastic changes in cleaved caspase-3 protein levels in the total cell population (Figure 3 and Figure 5), coupled with virus-induced protection from high ROS levels in all cells analysed (Figure 4). We reasoned that this could be explained by bystander effects, where infected monocytes might secrete factors which also result in e.g., upregulation of SOD2 in uninfected bystander cells, thereby protecting them from high ROS levels and apoptosis. To test this, we transferred supernatant from mock, WT or Δβ2.7 infected monocytes to uninfected, naïve monocytes, and 3 days post treatment, levels of SOD2 were assessed by western blot analysis. Figure 6E shows that supernatants from WT infected cells, but not cells infected with Δβ2.7 virus, strongly upregulated SOD2 in naïve CD14+ monocytes, indicating the involvement of secreted factor(s) in β2.7-dependent SOD2 upregulation. To rule out that this effect was a result of residual infectious virus in the infected cell supernatants, we UV-inactivated the supernatants prior to analysis to kill any residual progeny virus. We observed that the UV-inactivated supernatant still induced an increase in SOD2 levels (Figure 6E).

## 4. Discussion

β2.7 is an HCMV-encoded long non-coding RNA that is the most abundant latent viral transcript. Despite this, its role during latency remains has remained undefined and no latency-associated function for it has ever been described. We now show that β2.7 protects infected monocytes from apoptosis by preventing increases in ROS levels and by upregulating expression of the antioxidant enzymes, SOD2 and GPX-1. 

Reactive oxygen species pose a significant problem for all cells: whilst at low levels, they have important roles in cell signalling, high levels of ROS lead to irreversible oxidation of proteins and DNA, lipid peroxidation, inhibition of MMP and ATP production, and, ultimately, apoptosis or necrosis. 

In this paper, we have observed that HCMV protects against ROS during infection of monocytes and prevents apoptosis of the infected host cell, likely optimising maintenance of the latent reservoir to enable efficient reactivation and dissemination of reactivated virus. 

We recognise that in our analyses, we generated ROS artificially with cadmium chloride in order to address the ability of HCMV to combat ROS induction. However, infected monocytes are likely to face many natural ROS-inducing insults during latent carriage in vivo. Firstly, monocytes are drawn to sites of inflammation, in which ROS levels are high [50]. Proinflammatory cytokines such as TNFα and IL-1β, likely to be abundant in such environments, elicit mitochondrial ROS formation [51]. Mitochondrial ROS in turn can drive higher TNFα expression [52], resulting in a positive feedback loop for ROS production [53]. Secondly, high ROS levels are involved in the differentiation of monocytes to macrophages or dendritic cells [54,55,56,57], and can also increase after periods of hypoxia when blood flow is restored (ischaemia/reperfusion injury) (reviewed in [58,59]). 

Furthermore, we hypothesize that latent infection itself raises ROS levels in the cell, given that we saw higher ROS levels in Δβ2.7 infected monocytes compared to mock infected, even in the absence of cadmium ion treatment. Indeed, binding of HCMV virions to cells has been shown to generate ROS [60], and it is also possible that latent HCMV alters mitochondrial metabolism as it does in lytic infection [61,62], resulting in higher ROS generation that needs to be corrected for.

In this study, we found that as well as protecting infected monocytes from high ROS levels, β2.7 also prevents high ROS levels and upregulates SOD2 in uninfected bystander cells by means of a secreted factor. Preventing ROS-induced cell death in surrounding cells may indirectly protect the infected cell; death of myeloid cells is thought to mediate further local inflammation and associated tissue damage [54]. Furthermore, although this paper has focussed on ROS-induced apoptosis, mitochondrial ROS are also important intracellular signalling molecules, and can activate JNK and p38 MAPK, resulting in proinflammatory cytokine IL-6 and TNFα secretion in monocytes [52]. Additionally, mitochondrial ROS activate the inflammasome in monocytes via redox sensitive TBP-2, resulting in the secretion of IL-1β and IL-18 [50,63,64,65]. Therefore, dampening ROS in surrounding cells may also prevent local production of proinflammatory cytokines. As well as protecting cells from inflammatory damage, this could also contribute to immune evasion by the infected monocyte. It would be interesting to test whether the supernatant from infected monocytes can additionally upregulate SOD2 in other types of cells which could be uninfected bystanders in vivo. 

We are, as yet, uncertain of the identity of the secreted factor responsible for SOD2 upregulation. SOD2 is often upregulated by proinflammatory cytokines [66], however, this seems incompatible with latent infection which overall simulates an anti-inflammatory microenvironment [67,68]. At this time, it also remains unclear how β2.7 may stimulate production of the secreted activator of SOD2, although it is well established that lncRNAs can modulate gene expression in a number of ways: they can bind promoters and recruit transcription factors, they can bind to multiple miRNAs and prevent them from degrading their target or can bind and sequester proteins away from their targets (reviewed in [69,70]). 

Our analysis has focussed on mitochondrial ROS: the ROS dye used herein (MitoSOX) localises to the mitochondria, as does one of the antioxidants we used to protect against cadmium toxicity (MitoTEMPO). However, other sources of ROS exist within monocytes. For example, inflammation can drive neutrophils and monocytes to produce extracellular ROS via NADPH oxidase 2 (NOX2), which can diffuse back into cells [53]. Monocytes are thought to be particularly susceptible to cell death induced by this source of ROS [71]. Whilst SOD2 is localised in the mitochondria and limited to dealing with mitochondrial superoxide, we also found that HCMV significantly upregulated another antioxidant enzyme, GPX-1, which is mostly cytoplasmic, and, therefore, could mitigate these other sources of ROS [72]. 

Finally, whilst it is known that expression of β2.7 in isolation can protect cells from rotenone-induced apoptosis during permissive infection [14], it is not known if this is also the case in undifferentiated myeloid cells. At present, our data are consistent with β2.7 acting directly in an antiapoptotic role. However, we accept that β2.7 expression, or lack of expression, in myeloid cells upon infection could result in changes in expression of other viral genes involved in managing oxidative stress, mitochondrial function or apoptosis in infected myeloid cells.

In conclusion, we have shown that the HCMV long non-coding RNA, β2.7, which is the most abundant latent viral transcript but previously had no defined role during latency, protects against ROS-induced apoptosis and upregulates antioxidant enzymes in infected and bystander monocytes. This function of β2.7 likely plays a key role in the maintenance and dissemination of latent HCMV infection in the host.

## Figures and Tables

**Figure 1 viruses-14-00246-f001:**
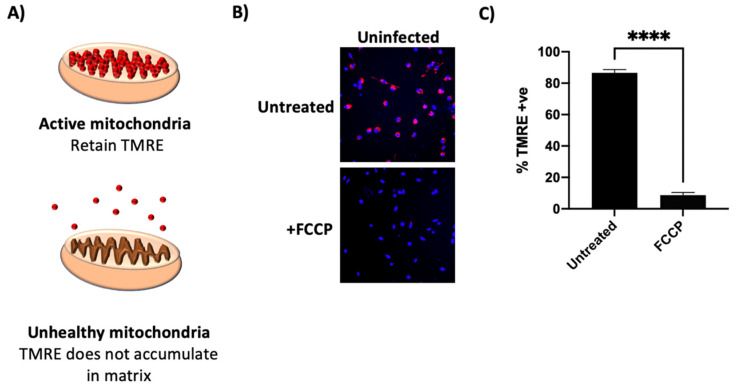
TMRE measures mitochondrial membrane potential (MMP) in monocytes. (**A**) TMRE, a fluorescent, positively charged stain, accumulates in actively respiring mitochondria because the mitochondrial matrix is negatively charged. When the mitochondrial membrane potential is dissipated, TMRE disperses. (**B**) Primary CD14+ monocytes isolated from peripheral blood were left untreated or treated with 20 μM FCCP, an uncoupler, for 10 min. Cells were then TMRE and Hoechst stained and photographed on a fluorescence microscope. Graph in (**C**) shows mean % TMRE + ve cells from samples analysed in triplicate. Significance was determined by a two-tailed Student’s *t*-test. **** = *p* < 0.0001.

**Figure 2 viruses-14-00246-f002:**
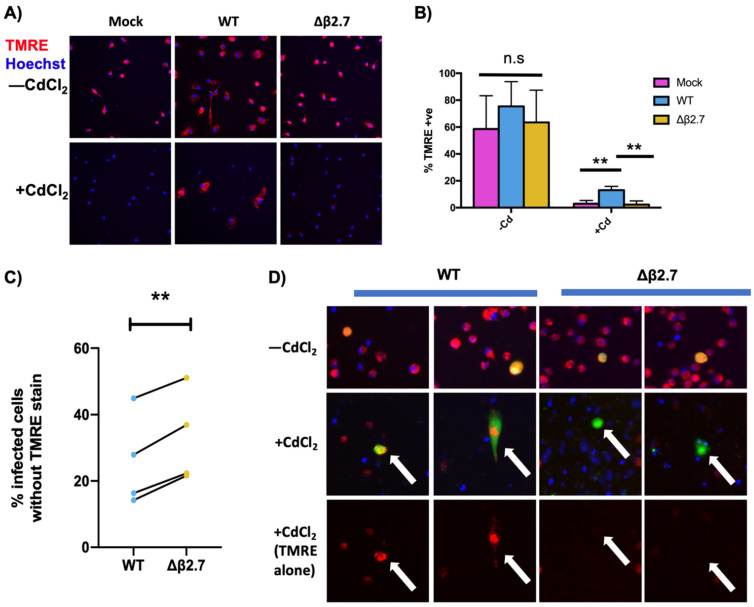
β2.7 protects against cadmium chloride-induced mitochondrial stress in HCMV infected CD14+ monocytes. (**A**,**B**) CD14+ monocytes were mock infected or infected with a WT or Δβ2.7 HCMV Toledo virus. At 14 d.p.i, half the cells were treated with 50 μM cadmium chloride for 24 h. All cells were then TMRE and Hoechst stained and photographed on a fluorescence microscope in (**A**). Graph in (**B**) shows mean % of TMRE + ve cells over three independent biological repeats. Error bars show standard deviation. Significant difference between mock and WT or WT and Δβ2.7 was determined using a one-way ANOVA with Tukey post hoc analysis. (**C**) CD14+ monocytes were infected with WT or Δβ2.7 HCMV TB40/E-SV40-GFP. 2 d.p.i, cells were treated with 50 μM CdCl_2_ for 24 h and then TMRE and Hoechst stained. GFP + ve cells with and without visible TMRE staining were enumerated. Graph shows the mean from 4 independent biological repeats from different blood donors. Statistical significance was determined by two-tailed paired *t*-test. ** = *p* <0.01. (**D**) Representative photos of CD14+ monocytes infected with WT or Δβ2.7 TB40/E-SV40-GFP that were treated with CdCl_2_ for 24 h at 4 d.p.i, and then TMRE and Hoechst stained. n.s. = not significant.

**Figure 3 viruses-14-00246-f003:**
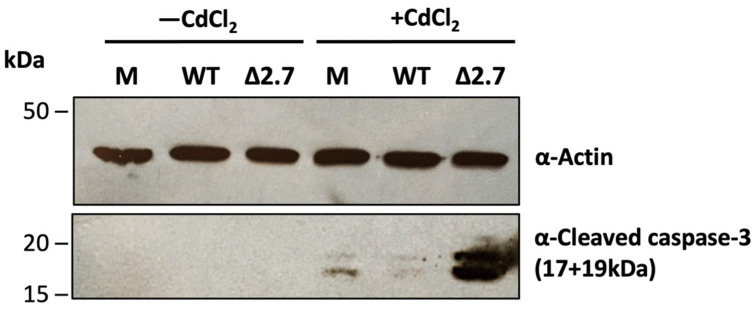
β2.7 protects against cadmium-induced apoptosis in infected monocytes. CD14+ monocytes were mock, WT or Δβ2.7 HCMV Toledo infected and at 6 d.p.i were treated with 50 μM CdCl_2_ for 24 h. Protein lysates from these cells were assessed for levels of cleaved caspase-3 and the loading control actin by immunoblot.

**Figure 4 viruses-14-00246-f004:**
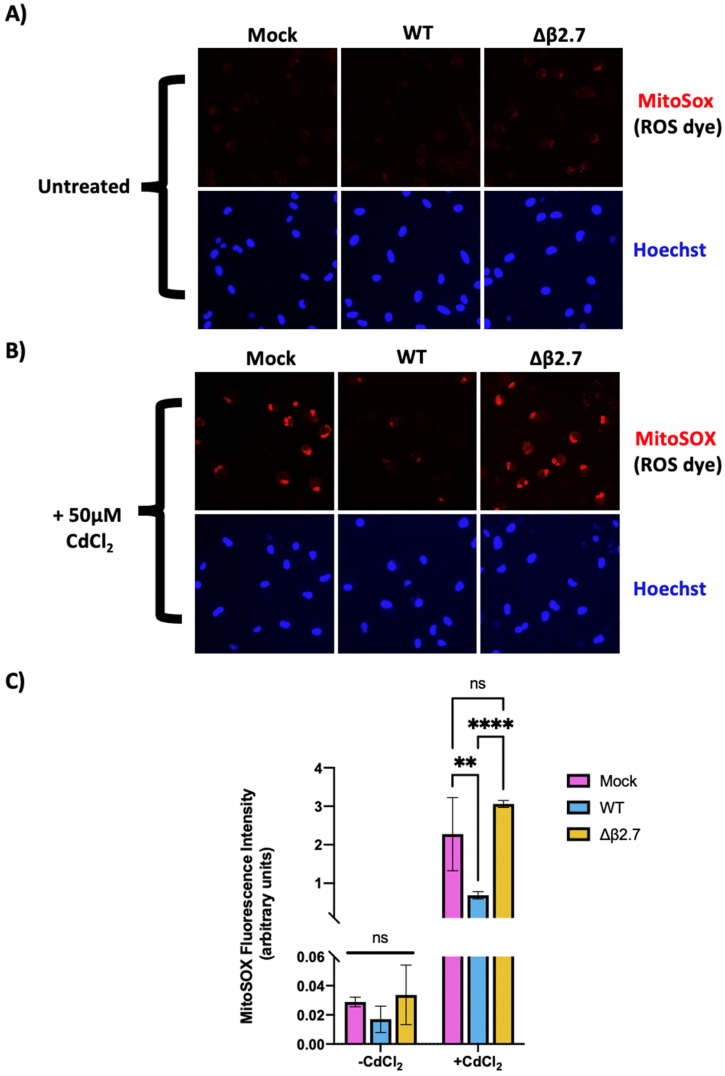
β2.7 protects against high ROS levels in infected monocytes. Primary CD14+ monocytes were isolated from peripheral blood and either mock infected or infected with WT or Δβ2.7 HCMV Toledo. 6 d.p.i, cells were untreated (**A**) or treated with 50 μM CdCl_2_ (**B**) for 24 h. Cells were then stained with Hoechst and the superoxide dye, MitoSOX, and imaged by fluorescence microscopy. (**C**) Quantification of MitoSOX fluorescence/cell for experiments described in A and B, where graph shows mean and standard deviation for three replicate experiments. Significance was determined by a two-way ANOVA with Tukey post hoc testing. ns = not significant, ** = *p* < 0.01, and **** = *p* < 0.0001.

**Figure 5 viruses-14-00246-f005:**
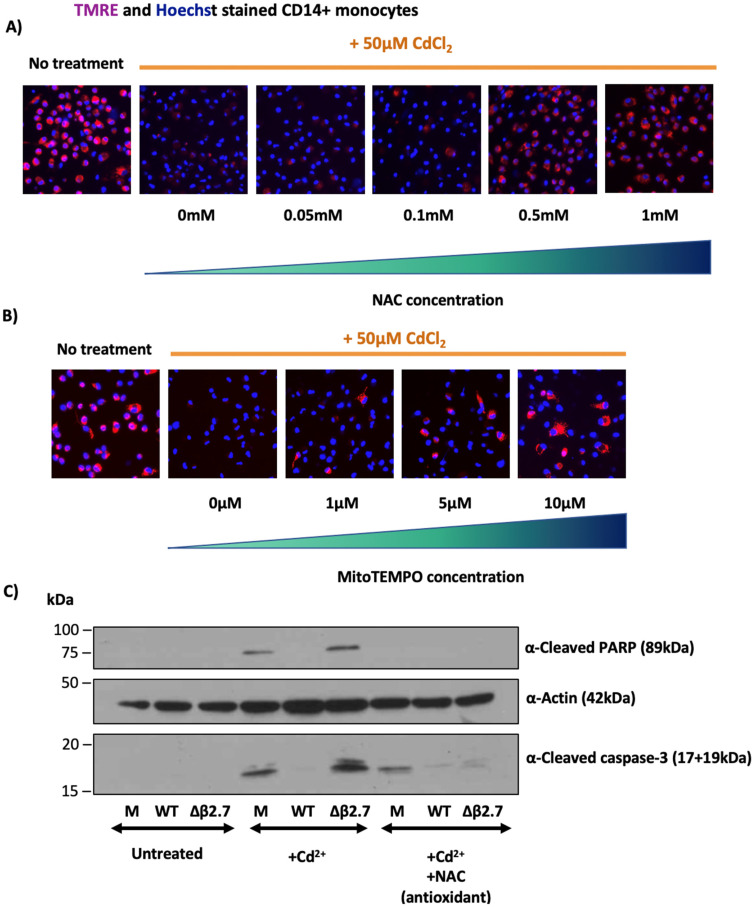
Lowering ROS levels with an antioxidant rescues Δβ2.7 virus from cadmium-induced apoptosis. (**A**,**B**) Primary CD14+ monocytes were treated with 50 μM cadmium chloride and increasing concentrations of the antioxidants (**A**) N-acetyl cysteine (NAC) or (**B**) MitoTEMPO. Then, 24 h post treatment, cells were stained with TMRE and Hoechst and imaged by fluorescence microscopy. (**C**) CD14+ monocytes were mock infected or infected with WT or Δβ2.7 HCMV Toledo. Next, 6 d.p.i, cells were treated with 50 μM CdCl_2_ with or without the antioxidant NAC. Finally, 24 h post treatment, protein lysates were harvested and immunoblotted for cleaved caspase-3, cleaved PARP and the loading control actin.

**Figure 6 viruses-14-00246-f006:**
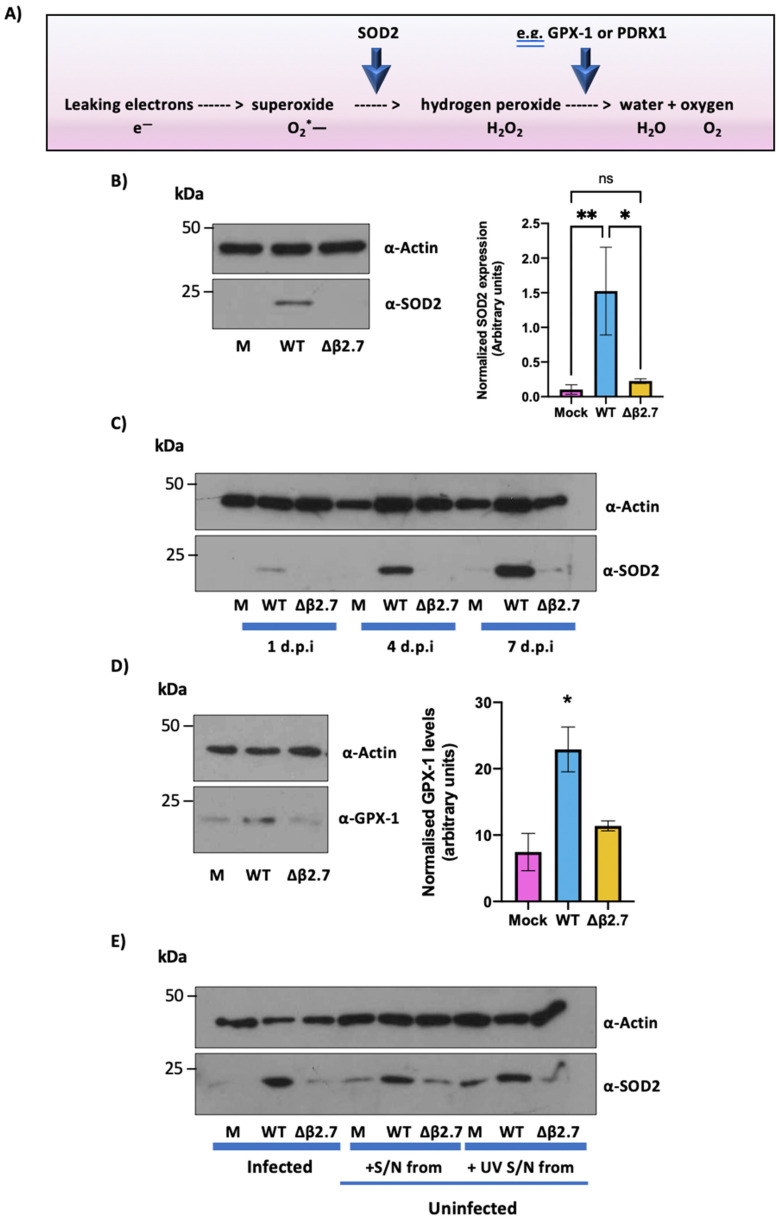
WT, but not Δβ2.7 infection upregulates SOD2 and GPX-1 in infected monocytes and bystander cells. (**A**) Electrons leaking from Complex I or III in the electron transport chain may react with oxygen to form superoxide. Superoxide is converted to hydrogen peroxide by the mitochondrial enzyme, SOD2, which is subsequently processed to water and oxygen by e.g., GPX-1 or PRDX1. (**B**) CD14+ monocytes were mock infected (M) or infected with WT or Δβ2.7 HCMV Toledo. 7 d.p.i, protein lysates from cells were harvested and assessed for SOD2 and actin levels by immunoblot. Graph shows densitometry from 3 independent biological repeats normalised to actin levels. Significance difference between Mock and WT and WT and Δβ2.7 was determined by one-way ANOVA with Tukey post hoc testing. ns = not significant, * = *p* < 0.05, and ** = *p* < 0.01. (**C**) Proteins from CD14+ monocytes infected with mock, WT or Δβ2.7 Toledo were harvested at 1, 4, and 7 d.p.i and immunoblotted for SOD2 and the loading control actin. (**D**) As described in (**B**), but protein lysates were immunoblotted for GPX-1 and actin. * = *p* < 0.05. (**E**) Supernatant from CD14+ monocytes that were mock, WT or Δβ2.7 Toledo infected was harvested at 7 d.p.i and either transferred directly (‘+S/N from’) or first UV inactivated (‘+UV S/N from’) and then placed onto uninfected CD14+ monocytes. Then, 3 days after treatment with supernatants, protein was harvested and immunoblotted for SOD2 and actin.

**Table 1 viruses-14-00246-t001:** Primer sequences used to generate Δβ2.7 TB40/E. Underlined sequence denotes sequence shared with galK.

Primer	Sequence
galK insertion F	CATCCCAAGCACTCCACACGCTATCACAGACCACGGACACGGCAAAAAATCCTGTTGACAATTAATCATCGGCA
galK insertion R	ACGTCTTTCCGCTTACTCAACGCGTCAGCCCGCGCTCGGCAGAGCTACCATCAGCACTGTCCTGCTCCTT
Reversion primer F	CAAGCACTCCACACGCTATCACAGACCACGGACACGGCAAAAAATTGGTAGCTCTGCCGAGCGCGGGCTGACGCGTTGAGTAAGCGGAAA
Reversion primer R	TTTCCGCTTACTCAACGCGTCAGCCCGCGCTCGGCAGAGCTACCAATTTTTTGCCGTGTCCGTGGTCTGTGATAGCGTGTGGAGTGCTTG

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
