# Peer review of "A Viral Long Non-Coding RNA Protects against Cell Death during Human Cytomegalovirus Infection of CD14+ Monocytes"

_viruses, 2022, doi:10.3390/v14020246_

Round 1
Reviewer 1 Report
The authors have responded adequately to my comments and concerns, and I am satisfied with their response. Appropriate changes have been made.
Reviewer 2 Report
While I would have welcomed a new experiment showing that B2.7 is sufficient for the phenotype, I accept the new paragraph in the Discussion that provides additional context for these findings.
This manuscript is a resubmission of an earlier submission. The following is a list of the peer review reports and author responses from that submission.
Round 1
Reviewer 1 Report
The manuscript entitled „A Viral Long Non-Coding RNA Protects Against Cell Death During Human Cytomegalovirus Latency“ by Perera et al. describes the infection of human peripheral blood-derived monocytes with WT HCMV and a virus lacking the β2.7 lncRNA (Δβ2.7). The authors found that monocytes inoculated with WT virus were protected from loss of mitochondrial membrane potential induced by CdCl2 whereas Δβ2.7-infected cells were not. Similarly, WT HCMV infection protected monocytes from ROS and apoptosis whereas Δβ2.7 infection did not. The authors further showed that SOD2 and GPX1 expression was upregulated in WT but not in Δβ2.7-inected monocytes and that this happens in both infected and uninfected bystander monocytes, suggesting a paracrine effect.
This is a very interesting study from the lab that first described the function of the β2.7 lncRNA in lytically infected cells. The present study furthers our understanding of how β2.7 lncRNA prevents cell death. The use of human monocytes, which support latent but not lytic infection, suggests that the described effect is operational in latently infected cells and bystander cells.
Questions and comments
- Throughout the paper, the authors stress that they analyzed latently-infected monocytes. However, I did not see any evidence that the infected cells were latently and not lytically infected. Is it possible that a certain percentage of the monocytes differentiated to macs and supported lytic replication? Did the authors do any control experiments to confirm that the cells, in which they observed protection from CdCl2 (Fig. 2), were latently infected? I assume that GFP is expressed during both latent and lytic infection. If the authors cannot provide evidence for latent infection, they might have to tone down their claims. They could talk about HCMV-infected (instead of latently-infected) monocytes and state that these are likely latently infected.
- In Fig. 6.E the authors show that soluble/secreted factors in the supernatant of WT HCMV-infected monocytes induce SOD2 upregulation. Are these factors produced only by infected monocytes? Is it possible that these factors were also present in the virus stocks (probably produced on fibroblasts) used to infect the monocytes? Please describe in Materials and Methods how virus stocks were made and whether the virus was purified to remove supernatant and debris. If the presence of the factor in virus stocks cannot be excluded, the conclusions need to be adjusted. In that case, the protection from mitochondrial dysfunction and apoptosis would have to be attributed to the unknown soluble factor whose production depends on β2.7, not to β2.7 RNA transcribed in latently infected monocytes.
- All WT HCMV-infected cells in Fig. 4.B appear to be protected from CdCl2, even though the authors state that only 10% of monocytes become latently infected (line 287). This protection is probably mediated by the soluble/secreted factor. Please comment on this in the manuscript.
- Throughout the manuscript, the authors talk about Hoechst staining without mentioning the number of the dye. There are several Hoechst dyes with different properties. Please describe the Hoechst dye, N-acetyl cysteine, and MitoTEMPO and their sources in the Materials and Methods part.
Reviewer 2 Report
Viral lncRNAs play important roles in regulating gene expression and cell fate. The HCMV lncRNA B2.7 is highly expressed in latent and lytic phases of infection, but its role remains poorly understood. Here, using a B2.7 KO virus, Perera, et al. demonstrate that B2.7 protects latently infected monocytes from ROS-induced apoptosis by upregulating antioxidant enzymes SOD2 and GPX-1. Interestingly, this mechanism of SOD2 upregulation appears to involve a soluble factor, as supernatants from HCMV latently infected cells could upregulate SOD2 in naive cells. While the precise mechanism of this SOD2 regulation remains obscure, this report provides a tantalizing starting point for future mechanistic studies of the soluble factor and downstream consequences.
The strength of this manuscript is that it is well written and logical and describes a fascinating phenomenon, and makes good use of a KO virus and other available tools for studying ROS, while zeroing in on SOD2/GPX-1 gene expression and providing interesting mechanistic detail. Indeed, the figures and legends are so clear that they can be initially understood without consulting the body text of the results (although the resolution of some of the figures could be improved). Overall, this represents a fascinating and potentially important contribution to the HCMV and lncRNA literature.
The primary weakness of the manuscript is that by relying exclusively on the KO virus, it does not definitively link B2.7 to the antiapoptotic phenotype. This could be addressed by ectopic expression of B2.7 in monocytes, to demonstrate that it is sufficient to mediate SOD2/GPX-1 upregulation and anti-apoptotic effects in these cells. Such experiments are important because it is not clear whether B2.7 deletion in HCMV causes additional changes in expression of other viral genes that could be implicated in managing oxidative stress, mitochondrial function or apoptosis. It is also unclear whether the B2.7 deletion has impacts on the efficiency of latency establishment in monocytes or rates of reactivation from latency, both of which could impact the fate of the monocytes. If some of this information exists in the literature, it should be clearly discussed and put in context of the current study. Regardless, the ectopic expression experiment is straightforward and could provide welcome support for the model.